# Targeted multi-omic analysis of human skin tissue identifies alterations of conventional and unconventional T cells associated with burn injury

Daniel R Labuz[1,2], Giavonni Lewis[3], Irma D Fleming[3], Callie M Thompson[3], Yan Zhai[3], Matthew A Firpo[3], Daniel T Leung[1,2]*

[1]Division of Infectious Disease, Department of Internal Medicine, University of Utah, Salt Lake City, United States; [2]Division of Microbiology & Immunology, Department of Pathology, University of Utah, Salt Lake City, United States; [3]Department of Surgery, School of Medicine, University of Utah, Salt Lake City, United States

**\*For correspondence:**
daniel.leung@utah.edu

**Competing interest:** The authors declare that no competing interests exist.

**Abstract** Burn injuries are a leading cause of unintentional injury, associated with a dysfunctional immune response and an increased risk of infections. Despite this, little is known about the role of T cells in human burn injury. In this study, we compared the activation and function of conventional T cells and unconventional T cell subsets in skin tissue from acute burn (within 7 days from initial injury), late phase burn (beyond 7 days from initial injury), and non-burn patients. We compared T cell functionality by a combination of flow cytometry and a multi-omic single-cell approach with targeted transcriptomics and protein expression. We found a significantly lower proportion of CD8 + T cells in burn skin compared to non-burn skin, with CD4 + T cells making up the bulk of the T cell population. Both conventional and unconventional burn tissue T cells show significantly higher IFN-γ and TNF-α levels after stimulation than non-burn skin T cells. In sorted T cells, clustering showed that burn tissue had significantly higher expression of homing receptors CCR7, S1PR1, and SELL compared to non-burn skin. In unconventional T cells, including mucosal-associated invariant T (MAIT) and γδ T cells, we see significantly higher expression of cytotoxic molecules GZMB, PRF1, and GZMK. Multi-omics analysis of conventional T cells suggests a shift from tissue-resident T cells in non-burn tissue to a circulating T cell phenotype in burn tissue. In conclusion, by examining skin tissue from burn patients, our results suggest that T cells in burn tissue have a pro-inflammatory rather than a homeostatic tissue-resident phenotype, and that unconventional T cells have a higher cytotoxic capacity. Our findings have the potential to inform the development of novel treatment strategies for burns.

## Editor's evaluation

The work in analyzing the T cell repertoire by multi-omics analysis is very valuable for the field of wound biology and provides convincing data with regard to both conventional and conventional T cells and their putative contributions. This moves the field beyond examining classical mediators of wound healing such as macrophages and neutrophils. We look forward to seeing this important work in *eLife*.

## Introduction

Burn injuries are a major health issue globally, ranking among the leading causes of unintentional injury, and non-surgical treatment options are mostly limited to supportive clinical care (*Peck, 2011*; *Boldeanu et al., 2020*). Major burn trauma results in dysfunctional immune responses and is associated with an increased risk of infections leading to poor survival outcomes (*Moins-Teisserenc et al., 2021*; *Bergquist et al., 2019*; *Jeschke et al., 2014*). Serum levels of interleukin (IL)–6, IL-10, tumor necrosis factor (TNF)-α, and interferon (IFN)-γ are significantly higher in burn trauma patients compared to non-burn (*Mace et al., 2012*; *Finnerty et al., 2007*). Similarly, higher serum concentrations of IL-6, IL-8, IL-10, GM-CSF, IL-12p70, IFN-γ, and TNF-α, are associated with sepsis and death in children with severe burns (*Finnerty et al., 2007*). Despite the many studies in this area that examine immune responses in blood to understand the systemic effects of burn injury, there is a paucity of studies examining the local immune response of injured burn tissue in humans.

Most reports characterizing T cells in the skin after a burn injury have been in mice (*Daniel et al., 2007*; *Rani et al., 2015*; *Sasaki et al., 2011*; *Rani and Schwacha, 2017*). In particular, αβ T cells and gamma-delta (γδ) T cells show increased production of cytokines TNF-α, IL-10, and IFN-γ in the days following burn injury (*Rani et al., 2014*). A significant number of αβ T cells and γδ T cells infiltrate the burn site, causing a total decrease in CD69 + cells, a marker of tissue residency in T cells (*Rani et al., 2015*; *Schenkel and Masopust, 2014*). Knockout of γδ T cells results in significantly higher TNF-α levels in the skin of mice following burn injury, suggesting a major role of γδ T cells in negating a pro-inflammatory environment (*Daniel et al., 2007*). However, given the discrepancy in the proportion of skin-resident γδ T cells between mice (where they account for >50% of total T cells) and humans (accounting for 2–5% of total T cells, *Ebert et al., 2006*; *Mestas and Hughes, 2004*), the clinical significance of the above findings is yet to be determined (*Schwacha, 2009*). In addition, while circulating T cells following burn injury in humans have elevated activation markers such as CD25 and HLA-DR (*Moins-Teisserenc et al., 2021*), the role of local or skin-resident T cells in human burn injury and burn-associated wound healing has not been well-studied.

Tissue-resident memory T cells ($T_{RM}$) in normal skin are effector memory cells that can persist for several years following an infection (*Clark et al., 2006*; *Zaid et al., 2014*). Thus, upon activation, skin $T_{RM}$ are potent producers of cytokines IFN-γ, IL-10, and TNF-α (*Clark et al., 2006*; *Clark et al., 2012*). Skin $T_{RM}$ are characterized by their high expression of CD103 (encoded by ITGAE) and CD69, and low expression of S1PR1, which allows them to persist in the skin by inhibiting the egress and circulation of cells to skin draining lymph nodes (*Clark, 2015*; *Shiow et al., 2006*). In comparison, skin central-memory T cells ($T_{CM}$), which recirculate between blood and ski (*Clark et al., 2012*), express CCR7 and CD62L (encoded by SELL). The impact of acute inflammatory insults such as burns on $T_{RM}$ and $T_{CM}$ populations remains unknown.

Mucosal-associated invariant T (MAIT) cells, a recently discovered unconventional innate-like T cell subset, have been shown to have capacity to participate in tissue repair (*Constantinides et al., 2019*; *Hinks et al., 2019*; *Leng et al., 2019*; *Lamichhane et al., 2019*). MAIT cells express a semi-invariant T cell receptor (TCR)-α consisting primarily of TRAV1-2 in humans and a variant but restricted variety of β chains. MAIT cells can be activated through a TCR-dependent pathway through presentation of a microbial-derived riboflavin metabolite, 5-(2-oxopropylideneamino)-6-d-ribitylaminouracil (5-OP-RU), by MHC class-1 related protein (MR1) on antigen-presenting cells (APC) (*Kjer-Nielsen et al., 2012*). MAIT cells can also be activated through a TCR-independent manner through cytokines IL-12, IL-15, and IL-18 (*van Wilgenburg et al., 2016*). Upon activation, MAIT cells express a variety of cytokines, including IFN-γ, TNF-α, IL17A, and IL-22 (*Gibbs et al., 2017*). The type of cytokine expressed is primarily influenced by the environment with peripheral blood MAIT cells expressing pro-inflammatory cytokines IFN-γ and TNF-α and tissue-resident MAIT cells expressing IL-17A and IL-22 (*Gibbs et al., 2017*). The wound healing capacity of human MAIT cells has been hypothesized to be a function of cytokine expression involving IL-17A and IL-22 pathways (*Constantinides et al., 2019*; *Leng et al., 2019*). In mice, it was shown that skin MAIT cells, when activated with a topical 5-OP-RU MAIT ligand, can promote tissue repair through an IL-17A pathway (*Constantinides et al., 2019*). Despite their suggested role in tissue repair, the activity of MAIT cells in burn tissue has yet to be determined.

Given the knowledge gap around the role of T cells in human burn injury, our primary objective was to examine the landscape of conventional and unconventional T cells following burn injury. We used flow cytometry to analyze the cytokine profiles of conventional T cells and unconventional T

cells including MAIT and γδ T cells. We compared populations of CD3 + cells within non-burn and burn tissue using a targeted T-cell panel with single-cell RNA sequencing (scRNA-seq) to determine specific subsets of T cells within the tissue and their gene expression profiles. We found that conventional CD3 + T cells and MAIT cells in burn-injured skin tissue produce higher levels of critical pro-inflammatory cytokines than in non-burn tissue. We also found that burn tissue T cell populations express higher levels of homing receptors associated with tissue residency, including S1PR1, SELL, and CCR7 compared to non-burn tissue T cells.

## Results

### Conventional CD8+ T cell populations are lower and conventional CD4+ T cell populations are higher in burn tissue compared to non-burn tissue

We used flow cytometry to determine the phenotype and frequency of T cell populations in burn tissue (12 samples, 11 subjects, 11.0% ± 7.4% TBSA), and non-burn tissue (7 samples, 7 subjects, 10.4%±4.7 TBSA; *Figure 1A*, *Table 1*, *Figure 1—figure supplement 2A*). Acute burn tissue samples (discarded tissue via tangential excision) were collected from patients within 7 days of initial injury, while late phase burn tissue was collected beyond 7 days of initial injury. Five non-burn tissue samples were collected from discarded split thickness autografts of acute burn patients (labeled 'acute non-burn' in *Table 1*), whereas two non-burn tissue samples were from patients who were undergoing surgeries not related to acute burn injuries (labeled 'non-burn' in *Table 1*). CD3 + T cells' frequency as proportion of live cells (*Figure 1B and C*) and total cell counts (*Figure 1D*), were not significantly different in any tissue comparison. We found a significantly higher proportion of CD4 + T cells in burn tissue (median 62.3% [interquartile range, IQR, 46.2%–70.9%]) compared to non-burn tissue (35.4% [18.6%–41.2%]; *Figure 1E*, p=0.02), while we saw lower proportion of CD8 + T cells in burn tissue (23.0% [18.9%–26.0%], p<0.0001) compared to non-burn tissue (53.4% [44.5%–69.6%]) (*Figure 1F*). We did not see any significant differences in absolute numbers of CD4 + or CD8+ T cells between burn and non-burn skin. (*Figure 1—figure supplement 2B, C*). In unconventional CD3 + T cell populations, there were no significant differences in proportions of Vα24-Jα18+ iNKT cells (*Figure 1G and H*), TCRγδ+ T cells (*Figure 1G1*), and MR1-5-OP-RU-tetramer+TRAV1-2+ MAIT cells (*Figure 1J and K*), between burn tissue and non-burn tissue. We analyzed separately the 'acute burn' and 'acute non-burn' groups, and showed similar differences between these groups for both CD8 +and CD4+ populations (*Figure 1—figure supplement 1*). Taken together, we found a markedly lower proportion of conventional CD8 + T cells and a higher proportion of CD4 + T cells in burn tissue compared to non-burn tissue, but no differences were seen in the proportions of unconventional T cells.

### Conventional CD4 + and CD8+ T cells in burn tissue show lower CD69 expression and produce more IFN-γ upon stimulation compared to T cells from non-burn tissue

Focusing on conventional T cells using flow cytometry, we found that burn tissue CD4 + T cells showed significantly lower expression of the T cell residency marker CD69 (p=0.0002) (*Figure 2A and B*, *Mackay et al., 2013*; *Gebhardt et al., 2009*; *Kumar et al., 2017*) and significantly higher expression of CD38, a marker of chronic activation (p=0.002) compared to non-burn CD4 +T cells (*Figure 2A and B*; *Song et al., 2020*). We then looked at the expression of specific intracellular cytokines suggested by prior studies to be important in burn tissue immune environment (*Blears et al., 2020*). After treatment for 2 hr with PMA-ionomycin, we saw significantly higher expression of IFN-γ (p=0.0019) and TNF-α (p=0.0068) in CD4 + T cells in burn tissue compared to non-burn tissue (*Figure 2C–E*). Lastly, we saw significantly higher proportion of IL-10 +CD4+ T cells in unstimulated burn tissue compared to unstimulated non-burn tissue (p=0.017, *Figure 2F*). We had similar findings in CD8 + T cells, where CD69 expression was significantly lower (p<0.0001) and CD38 significantly higher (p=0.015) in burn tissue and compared to non-burn tissue (*Figure 2G and H*), and upon stimulation, we saw in burn tissue higher expression of IFN-γ (p=0.0003) and TNF-α (p=0.0002) (*Figure 2J–L*) compared to non-burn tissue. We did not see any differences in IL-10 production or pro-inflammatory cytokine production in unstimulated conditions in CD8 + T cells from burn tissue (*Figure 2I–L*). When comparing 'acute burn' and 'acute non-burn' we saw similar differences in CD69 expression and pro-inflammatory

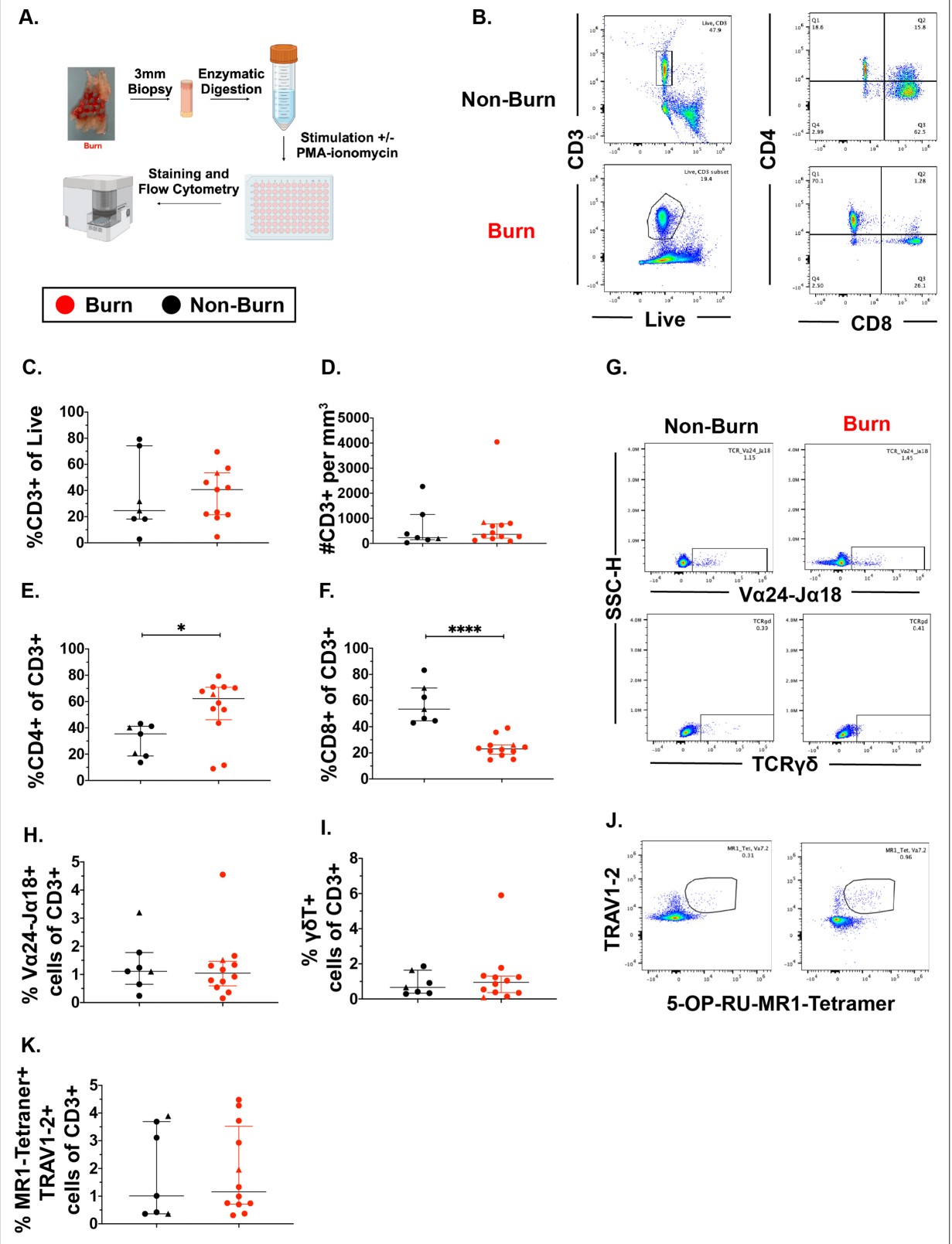

**Figure 1.** Conventional CD8 +T cell populations are lower and CD4 +T cells populations are higher in burn tissue compared to non-burn tissue. Overview of the processing and analysis of T cells in burn and non-burn skin tissue for flow cytometry (**A**). Representative gating of CD3+, CD4 + and CD8+ T cells from live cells from non-burn tissue and burn tissue (**B**). Frequency of CD3 + T cells from live cells in burn or non-burn tissue measured by flow cytometry (**C**). Absolute count of CD3 + T cells from specific volume of burn or non-burn tissue measured by flow cytometry (**D**). Frequency of

*Figure 1 continued on next page*

*Figure 1 continued*

CD8 + and CD4+ T cells by flow cytometry (**E–F**). Frequency and gating strategy of unconventional T cell populations: iNKT (**G,H**), γδ T cells (**G,I**), or MAIT cells (**J,K**). Error bars shown are of median with interquartile range. Differences between burn and non-burn were calculated using Mann-Whitney *U* test with *p<0.05, ** p<0.01, ****p<0.0001.

The online version of this article includes the following figure supplement(s) for figure 1:

**Figure supplement 1.** Conventional CD8 + T cell populations are lower and CD4 + T cells populations are higher in acute burn tissue compared to acute non-burn tissue.

**Figure supplement 2.** Conventional CD4 + and CD8+ T cell absolute numbers are not significantly different between burn and non-burn tissue.

cytokine output in CD4 +and CD8+T cells following stimulation (**Figure 2—figure supplement 1**). Taken together, we found that conventional CD4 + and CD8+ T cells from burn tissue have lower CD69 expression and produce higher pro-inflammatory cytokines IFN-γ and TNF-α upon stimulation compared to non-burn tissue.

## Unconventional T cells in burn tissue have lower CD69 expression and produce more IFN-γ and TNF-α upon stimulation compared to non-burn tissue

We then examined unconventional T cells, including γδ T cells and MAIT cells using flow cytometry. As seen in conventional T cells, CD69 expression in MAIT cells was significantly lower (p=0.0002) in burn tissue than non-burn tissue (**Figure 3A and B**). Burn tissue MAIT cells had significantly higher CD38

**Table 1.** Demographics of patients used in this study.

n/a implies the skin sample came from a patient who was not a severe burn patient or several years past initial burn injury.

| Age group | Sex | Type | TBSA | Days sample taken after burn | Type of injury | Category |
|---|---|---|---|---|---|---|
| 70–79 | M | Burn | 16 | 4 | Flame | Acute Burn |
| 30–39 | M | Burn | 9 | 11 | Scald | Late Phase Burn |
| 60–69 | M | Burn | 2 | 4 | Electrical | Acute Burn |
| 20–29 | M | Burn | 10 | 6 | Flame | Acute Burn |
| 40–49 | M | Burn | 13 | 7 | Flash | Acute Burn |
| 10–19 | F | Burn* | 2 | 25 | Flame | Late Phase Burn |
| 50–59 | M | Burn | 14 | 5 | Flame | Acute Burn |
| 5–9 | F | Burn* | 11 | 9 | Scald | Late Phase Burn |
| 40–49 | M | Burn* | 26 | 7 | Scald | Acute Burn |
| 40–49 | M | Burn | 16 | 11 | Flame | Late Phase Burn |
| 30–39 | F | Burn | 1 | 27 | Contact | Late Phase Burn |
| 40–49 | M | Non-Burn | 9 | 9 | Flame | Acute Non-Burn |
| 30–39 | F | Non-Burn[†] | 10 | n/a | Burn scar | Non-Burn |
| 30–39 | M | Non-Burn | 0 | n/a | Necrotizing fasciitis | Non-Burn |
| 40–49 | F | Non-Burn* | 0 | n/a | Necrotizing fasciitis | Non-Burn |
| 40–49 | M | Non-Burn* | 26 | 14 | Scald | Acute Non-Burn |
| 30–39 | M | Non-Burn* | 13 | 16 | Electrical and flame | Acute Non-Burn |
| 40–39 | F | Non-Burn | 49 | 9 | Flame | Acute Non-Burn |
| 60–69 | M | Non-Burn | 2 | 4 | Electrical | Acute Non-Burn |

*Denotes sample was used for scRNA-seq. Acute non-burn = autograft tissue from acute burn patients.

[†]Denotes sample was used for data analysis in **Figure 1—figure supplements 1 and 2**, **Figure 2—figure supplement 1** only.

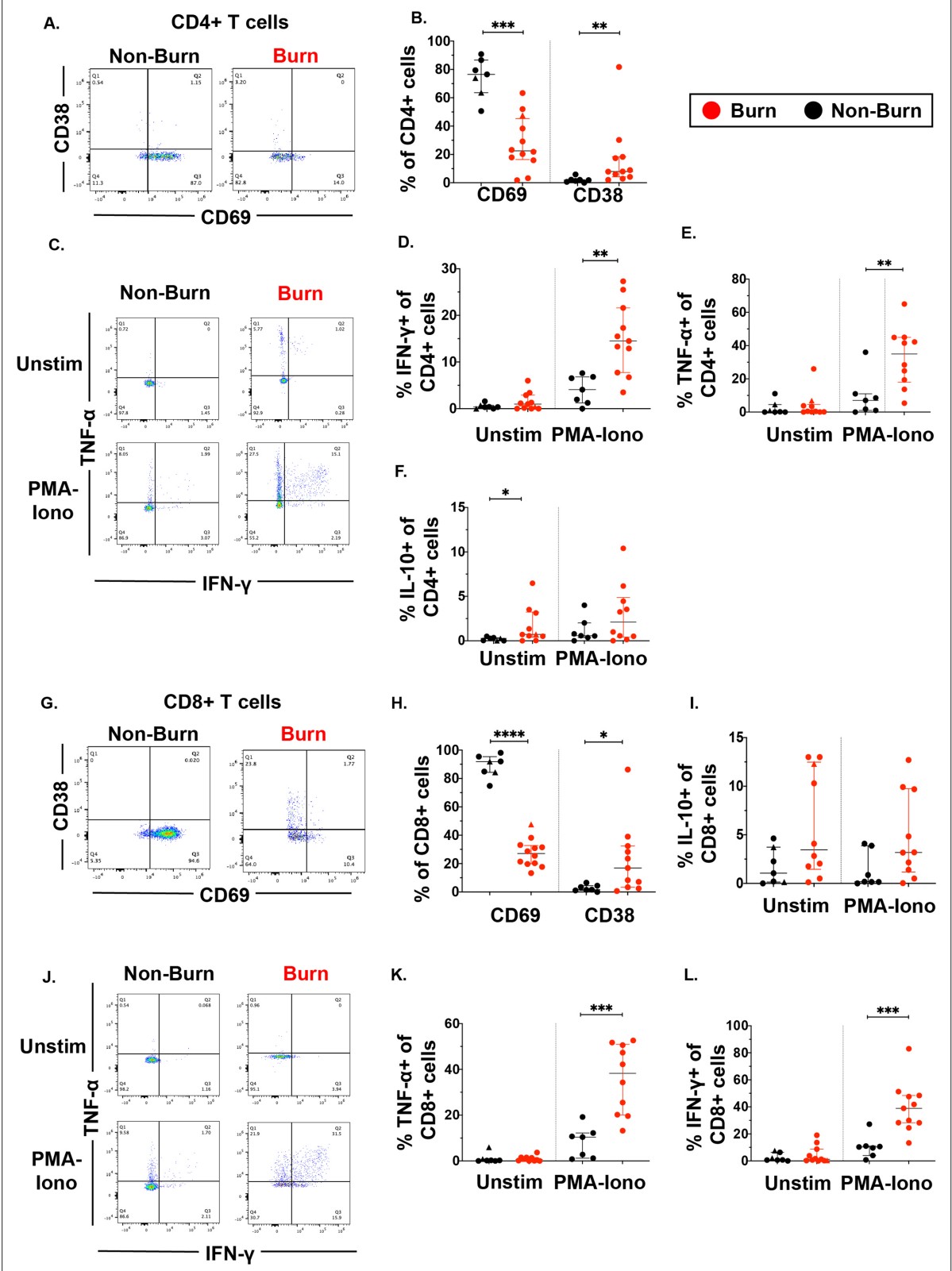

**Figure 2.** Conventional CD4 +and CD8+T cells in burn tissue produce more pro-inflammatory cytokines upon stimulation compared to those from non-burn tissue. Representative gating of CD69 and CD38 of CD4 + T cells and frequency of CD69 + and CD38+ CD4+ T cells in burn and non-burn tissue (**A,B**). Representative gating of IFN-γ+and TNF-α+ CD4+ T cells in unstimulated conditions and after 2 hr PMA-ionomycin stimulation in burn tissue and non-burn tissue (**C**). Quantification of frequency of IFN-γ+, TNF-α+, and IL-10 + CD4+T cells between burn and non-burn tissue in unstimulated

*Figure 2 continued on next page*

*Figure 2 continued*

conditions and after 2 hr PMA-ionomycin stimulation (**D–F**). Representative gating of CD69 and CD38 of CD8 + T cells and frequency of CD69 + and CD38+ CD8+T cells in burn and non-burn tissue (**G,H**). Representative gating of IFN-γ+ and TNF-α+ CD8+T cells in unstimulated conditions and after 2 hr PMA-ionomycin stimulation in burn tissue and non-burn tissue (**J**). Quantification of frequency of IFN-γ+, TNF-α+, and IL-10 + CD8+T cells between burn and non-burn tissue in unstimulated conditions and after 2 hr PMA-ionomycin stimulation (**I, K, L**). Error bars shown are of median with interquartile range. Differences between burn and non-burn were calculated using Mann-Whitney *U* test with *p<0.05, **p<0.01, ***p<0.001, ****p<0.0001.

The online version of this article includes the following figure supplement(s) for figure 2:

**Figure supplement 1.** Conventional CD4 +and CD8+T cells in acute burn tissue and late phase burn tissue produce more pro-inflammatory cytokines upon stimulation compared to those from acute non-burn tissue.

expression compared to non-burn tissue (p=0.01) (*Figure 3B*). With 2 hr PMA-Ionomycin stimulation, we saw a significantly higher expression of IFN-γ (p=0.0002) and TNF-α (p=0.003) in burn tissue MAIT cells compared to non-burn tissue MAIT cells (*Figure 3C–E*), but no differences were seen in IL-10 and IL-17A between these tissues (*Figure 3F and G*), nor in any examined cytokines in unstimulated conditions. In γδ T cells, there was significantly lower CD69 expression in burn tissue compared to non-burn tissue (p=0.006), although CD38 expression was not significantly different (*Figure 3H1*). Following PMA-Ionomycin stimulation, we saw significantly higher IFN-γ (p=0.01) and TNF-α (p=0.007) production in burn tissue γδ T cells compared to non-burn tissue (*Figure 3K and L*). In unstimulated conditions, significantly higher TNF-α (p=0.015) in burn tissue γδ T cells was noted compared to non-burn tissue, while there were no other significant differences in cytokine production (*Figure 3J–M*). When comparing 'acute burn' and 'acute non-burn' we saw similar differences in CD69 expression and IFN-γ and TNF-α output following stimulation in MAIT cells and γδ T cells (*Figure 3—figure supplement 1*). Overall, like conventional CD4 + and CD8+ T cells, MAIT cells and γδ T cells in burn tissue have a higher capacity to express the pro-inflammatory cytokines IFN-γ and TNF-α upon stimulation.

## Conventional T cells in burn tissue are transcriptionally distinct from those in non-burn tissue

To get a broader view of the effects of burn trauma on human skin CD3 +T cells, we performed targeted single-cell RNA-sequencing (scRNA-seq), using the BD Rhapsody platform, on three burn tissues and three non-burn tissue samples (*Table 1*). Dimensional reduction and visualization using Uniform Manifold Approximation and Projection (UMAP) demonstrated that non-burn and burn T cells cluster separately in fifteen different populations (*Figure 4A and B*). The differentially expressed genes in the largest cluster, cluster 0, which primarily consisted of non-burn T cells, include *IL7R* (log2 fold change comparing burn to non-burn tissue $L_2FC = 1.1$, p.adj <9e-330), *LGALS1* ($L_2FC = 1.1$, p.adj=2.8e-320) and *CD69* ($L_2FC = 1.3$, p.adj=6.6e-269) (*Figure 4C and D*). Cluster 1, a large population consisting mostly of burn tissue, had significantly higher expression of genes encoding for homing receptors compared to non-burn T cells, including *SELL* ($L_2FC = 2.8$, p.adj <9e-330) and *S1PR1* ($L_2FC = 2.9$, p.adj <9e-330) (*Figure 4C and D*). When we look at the top 8 genes differentiating burn from non-burn samples, we saw that T cells in burn tissue have higher expression of *SELL*, *CCR7* and *S1PR1* and have lower *CD69* expression compared to non-burn samples (*Figure 4E*). When comparing protein expression using antibody-oligonucleotide conjugates (AbSeq), we found lower protein expression of PD-1 and CD25 in the burn tissue T cells compared to non-burn (*Figure 4F*). We confirmed our previous findings from flow cytometry, finding a lower proportion of CD8 + T cells (39% vs 63% CD8A expression, p.adj=5.4e-103) in burn tissue (*Figure 4G*). We also confirmed the differences in gene expression of *ITGAE* (encoding CD103) and *CD69* by flow cytometry, finding that there are significantly lower frequencies of CD103 +CD69+CD3+ T cells in burn tissue compared to non-burn tissue (*Figure 4H*).

Lastly, we see that expression of effector molecules *IFNG*, *TNF*, *NAMPT*, *GZMB*, *GZMH*, and *PRF1* is lower in burn compared to non-burn CD3 + T cells (*Figure 4—figure supplement 1A-C*). Gene ontology (GO) analysis shows that T cell activation and positive regulation of cytokine production are positively associated with burn T cells while apoptotic processes are negatively associated with burn T cells. (*Figure 4—figure supplement 1D*). Lastly, KEGG analysis of CD3 + T cells shows Th17, Th1, and Th2 cell differentiation are positively associated with burn T cells while cytokine-cytokine

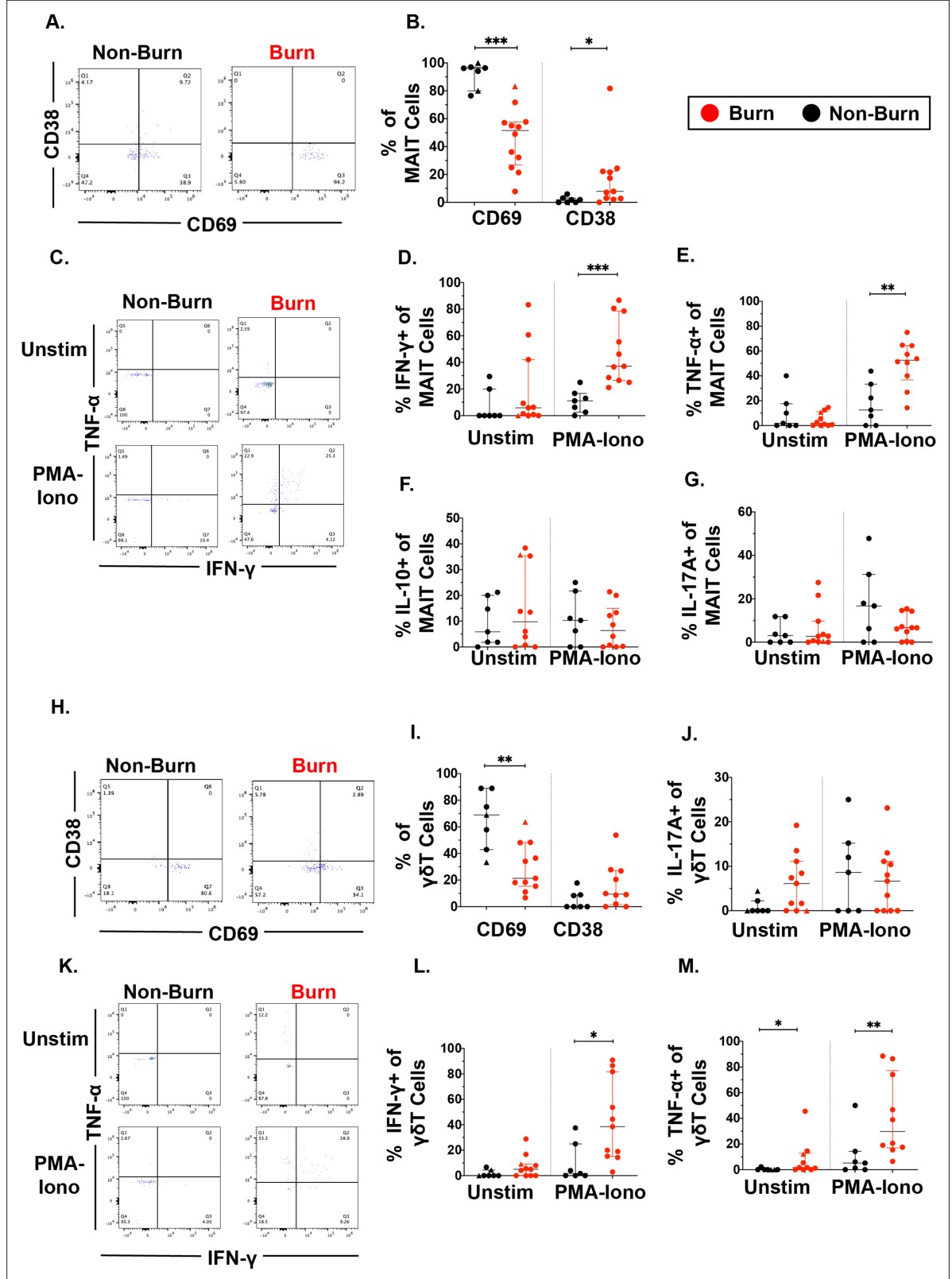

**Figure 3.** Unconventional T cells in burn tissue produce more IFN-γ and TNF-α upon stimulation compared to non-burn tissue. Representative gating of CD69 + and CD38+ MAIT cells in burn tissue and non-burn tissue (**A**) Frequency of CD69 +and CD38+ MAIT cells in burn tissue and non-burn tissue (**B**). Representative gating of IFN-γ+ and TNF-α+ of unstimulated MAIT cells and stimulated MAIT cells following 2 hr PMA-ionomycin stimulation in burn tissue and non-burn tissue (**C**). Frequency of IFN-γ+, TNF-α+, IL-10+, and IL-17A+ in unstimulated and stimulated MAIT cells after

*Figure 3 continued on next page*

Figure 3 continued

2 hr PMA-ionomycin in burn tissue and non-burn tissue (**E–G**). Representative gating of CD69 + and CD38+ γδ T cells and quantification of frequency of CD69 + and CD38+ γδ T cells in burn tissue and non-burn tissue (**H,I**). Frequency of IL-17A+ in unstimulated and stimulated γδ T cells after 2 hr PMA-ionomycin in burn tissue and non-burn tissue (**J**). Representative gating of IFN-γ+ and TNF-α+ of unstimulated γδ T cells and stimulated γδ T cells and quantification of frequency of IFN-γ+ and TNF-α+ γδ T cells in burn tissue and non-burn tissue (**K–M**). Error bars shown are of median with interquartile range. Differences between burn and non-burn were calculated using Mann-Whitney *U* test with *p<0.05, **p<0.01, ***p<0.001.

The online version of this article includes the following figure supplement(s) for figure 3:

**Figure supplement 1.** Unconventional T cells in burn acute tissue produce more IFN-γ and TNF-α upon stimulation compared to acute non-burn tissue.

receptor interactions and T cell receptor signaling pathways are negatively associated with burn T cells (*Figure 4—figure supplement 1E*).

On closer examination of conventional T cell subsets, we found that CD4 + T cells from burn and non-burn tissue comprised of 11 clusters and had distinctly separate clustering based on UMAP analysis (*Figure 5A and B*). Cluster 0, which consisted mostly of non-burn CD4 + T cells, has high expression of *IL7R*, *CD69*, and *LGALS1* suggesting a tissue residency signature (*Figure 5A–C*). Meanwhile, cluster 1, primarily burn CD4 + T cells, expressed high *S1PR1*, *CCR7*, and *SELL* levels (*Figure 5A–C*). Cluster 2, situated in non-burn tissue, highly expressed *FOXP3* and *CTLA4*, suggesting an enriched T$_{reg}$ population (*Figure 5A–C*). When we look at the highly differentially expressed receptors in CD4 + cells, we see that *S1PR1*, *CCR7*, *SELL*, and *ITGA4* were exclusively expressed in burn tissue (*Figure 5D*). Confirming our findings with flow cytometry, the AbSeq showed that CD4 + T cells in burn tissue express significantly lower CD69 and higher CD38, compared to non-burn tissue (*Figure 5E*). A closer look at effector molecules shows that burn CD4 + T cells have lower expression of *NAMPT*, *GZMA*, *PRF1*, *TNF*, and *CCL5* compared to non-burn CD4 + T cells (*Figure 5—figure supplement 1A, B*).

When we examined CD8 + T cells, we saw similar clustering as CD4 + T cells based on burn status (*Figure 5F and G*). The large cluster 0 situated in non-burn tissue show upregulation of several cytotoxic molecules, including *GZMK*, *GZMH*, and *GNLY* (*Figure 5F–H*). A similar cytotoxic CD8 + T cell cluster was seen in burn tissue found at cluster 4 (*Figure 5F and H*). We also saw that expression of *NAMPT, TNF, IFNG, CCL4, and IL32* were significantly lower in burn tissue CD8 + T cells compared to non-burn CD8 + T cells (*Figure 5—figure supplement 1C, D*). Similar to CD4 + T cells from the scRNA-seq data, we saw a burn tissue CD8 + population in clusters 3, 4, and 8, which highly expresses homing markers *SELL* and *S1PR1* (*Figure 5H*). Overall, we observed a significant higher expression of *ITGA4*, *SELL*, *CCR7*, *SELPLG*, and *S1PR1* and lower expression of *CD69* in burn tissue CD8 + T cells compared to non-burn tissue (*Figure 5I*). The AbSeq of CD8 + T cells shows a lower expression of PD1, CD25, and CD69 in CD8 + T cells in burn tissue and higher expression of CD38 in comparison to non-burn tissue (*Figure 5K*). A closer examination of effector molecules shows that burn CD8+ T cells have lower expression of *NAMPT*, *GZMB*, *GZMH*, *TNF*, and *IL32* and higher expression off *CCL5 and GZMA* compared to non-burn CD8+ T cells (*Figure 5—figure supplement 1C, D*).

## Unconventional T cells in burn tissue comprise a highly cytotoxic population not seen in non-burn tissue

Similar to our findings with conventional T cells, MAIT cells (defined in AbSeq as Vα7.2+and CD161+) from burn tissue clustered separately from those in non-burn tissue by UMAP analysis (*Figure 6A and B*). Cluster 0, positioned in burn tissue, showed highly upregulated cytotoxic genes, including *PRF1*, *GZMK*, and *NKG7* (*Figure 6B and C*). Cluster 1, positioned in non-burn tissue, exhibited upregulation of *CD4*, *ICOS*, and *TIGIT*, suggesting a MAIT T$_{reg}$ lineage; notably, we also identified *FOXP3* + MAIT cells in this cluster (*Figure 6B and D*; *Vorkas et al., 2022*). A further look into transcription factors associated with burn tissue MAIT cells showed a high proportion of *RORC*, while *TBX21* (encoding for T-bet) and *ZBTB16* (encoding for PLZF) had similar expression across tissues (*Figure 6D*). As expected, burn tissue MAIT cells had lower expression of *CD69*, and higher expression of *S1PR1* and *CCR7*, like our findings in conventional T cell populations, compared to non-burn tissue (*Figure 6E*). Notably, we also saw a significantly higher expression of *CXCR6* in burn tissue (*Figure 6E*), suggesting this may be a potential mechanism of MAIT cell trafficking. Taken together, burn tissue MAIT cells are highly cytotoxic and express markers for egress from tissue compared to non-burn MAIT cells.

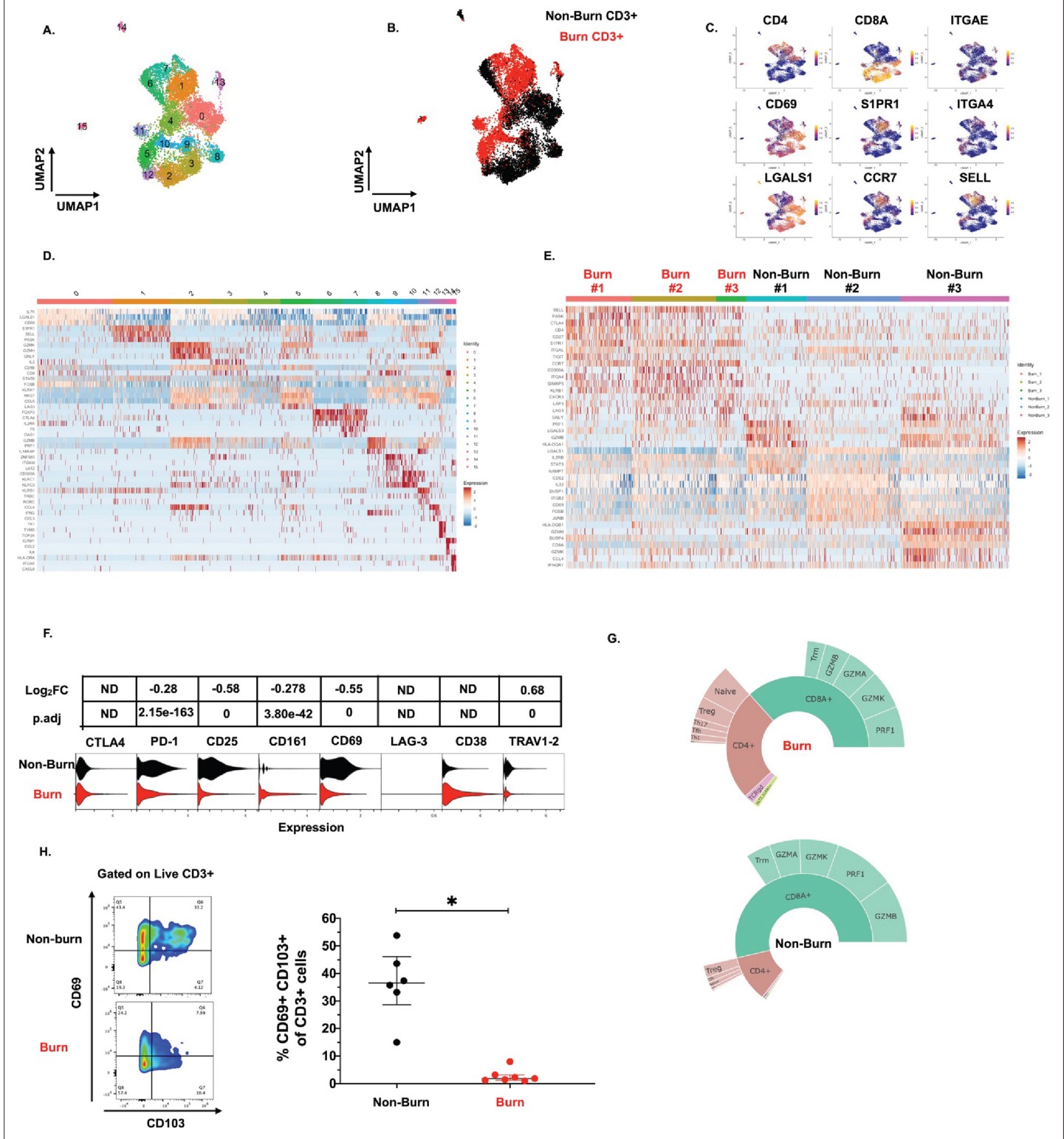

**Figure 4.** Conventional T cells in burn tissue are transcriptionally distinct from those in non-burn tissue. UMAP analysis using resolution = 0.8 in Seurat of 15 different clusters of T cells (**A**). Identification of burn and non-burn tissue T cells using Sample Tag calling from BD Rhapsody workflow (**B**). The key genes found that discriminate the 15 clusters (**C**). Heatmap of the top 5 differentially expressed genes using MAST that identify clusters associated with T cells in burn or non-burn tissue (**D**). Heatmap of the top 8 differentially expressed genes between the three burn and three non-burn samples (**E**). Violin plot of AbSeq expression of receptors between burn and non-burn CD3 + T cells (**F**). Sunplot of the proportion of T cell subsets of CD4+, CD8+, MAIT (TRAV1-2+and CD161+), and γδ T cells (TRDC +TRGC+) between burn and non-burn tissues, definitions of CD4 + and CD8+ subsets

*Figure 4 continued on next page*

*Figure 4 continued*

were taken from *Mair et al., 2020* (**G**). Flow cytometry showing gating strategy and frequency of CD69 +CD103+ of CD3+T cells in burn and non-burn tissue (**H**). Error bars shown are of median with interquartile range. Differences between burn and non-burn were calculated using Mann-Whitney *U* test with *p<0.05.

The online version of this article includes the following figure supplement(s) for figure 4:

**Figure supplement 1.** Differential expression of cytokines and effector molecules of conventional CD3 +T cells in burn and non-burn tissues.

We next looked at γδ T cells (defined as TRDC +and TRGC+), and we saw six distinct populations when analyzed using UMAP (*Figure 6F and H*). Cluster 0, mostly from burn tissue, exhibited high expression of cytotoxic molecules *PRF1* and *GZMB* (*Figure 6H*). Interestingly, cluster 0 also expressed a high level of *KRLB1* and *IL18RAP*, indicating a population of Vδ2 γδ T cells, known for being transcriptionally similar to MAIT cells (*Provine et al., 2018*; *Wragg et al., 2020*; *Gutierrez-Arcelus et al., 2019*) Meanwhile, cluster 1, mainly from non-burn γδ T cells showed high expression of *LGALS1* (*Figure 6F and H*). Clusters 0 and 2 in burn tissue express *SELL*, *CCR7* and *S1PR1*, genes previously upregulated in burn tissue CD4 + and CD8+ T cells (*Figure 6F and H*). Burn tissue γδ T cells had higher expression of *ITGAE* and *CXCR6* compared to non-burn tissue. (*Figure 6I*). When comparing AbSeq profiles, burn tissue γδ T cells show lower expression of CD25, CD69, and PD1 while having comparable CD38 and CTLA4 expression to non-burn γδ T cells (*Figure 6J*). Burn tissue γδ T cells show higher expression of CD161 compared to non-burn tissue γδ T cells (*Figure 6J*). Overall, burn tissue γδ T cells displayed upregulation of cytotoxic and homing genes compared to non-burn tissue.

## Discussion

In this report, we describe the phenotype and functionality of conventional T cells, MAIT cells, and γδ T cells in human non-burn skin and burn tissue using combination of flow cytometry and a targeted multi-omic analysis of protein and gene expression at the single-cell level. Despite studies reporting the effect of burn injury on circulating blood T cells (*Moins-Teisserenc et al., 2021*; *Hur et al., 2015*; *Sobouti et al., 2020*), little is known about T cells in burn skin tissue. We found that following stimulation, conventional and unconventional T cells in burn tissue produce more pro-inflammatory cytokines TNF-α and IFN-γ compared to non-burn tissue. Additionally, we found altered transcriptional expression of homing receptors among T cells in burn tissue, suggesting the migration of T cells from circulation, possibly accounting for differences in cytokine production capacity.

Prior studies of cytokines in blood have found that IL-6, IL-10, and TNF-α increase in burn compared to healthy pediatric patients, that elevated circulating TNF-α is associated with death, (*Sobouti et al., 2020*) and that decreased circulating TNF-α/IL-10 ratio is correlated with an increased chance of infection (*Tsurumi et al., 2016*). In addition, the concentration of circulating TNF-α in human blood peak in the first day of burn injury while IFN-γ peaks in days 2–5 of burn injury (*Bergquist et al., 2019*). Here, compared to non-burn tissue, we show that in burn tissue, a higher proportion of conventional T cells, γδ T cells, and MAIT cells produce TNF-α and IFN-γ upon stimulation. Based on studies showing the potential for TNF-α and IFN-γ to upregulate platelet-activating factor receptor on intestinal epithelial cells and promote wound closure (*Birkl et al., 2019*), it is possible that the increased capacity of burn tissue T cells to produce TNF-α and IFN-γ may contribute to responses against secondary infection and support wound healing.

We found a significantly lower frequency of conventional CD8 + T cells in burn tissue, with CD4 + cells making up the bulk of the conventional T cell population. In contrast to mouse models of burn injury that show a significant increase in total T cells or decreases in γδ T cells following burn injury (*Rani et al., 2015*; *Rani and Schwacha, 2017*), we did not find differences in total T cells or unconventional T cells in burn tissue. While our extraction time for burn tissue and TBSA were similar to those conventionally used in mouse models (extraction at 3–7 days post-burn and 25% TBSA achieved through scalding) (*Alexander et al., 2002*; *Schwacha and Somers, 1998*), the type of burn injury studied may complicate comparison of inflammatory responses between human and mice (*Jeschke et al., 2020*). Similarly, we saw a decrease in proportion of CD69 + conventional T cells in burn tissue compared to non-burn tissue, which others have reported in mice 3 days post burn (*Rani et al., 2015*). In addition, our finding that both acute and late phase burn tissue had increased IFN-γ+ conventional T cells compared to acute non-burn tissue matches studies in mice, where there is an increased

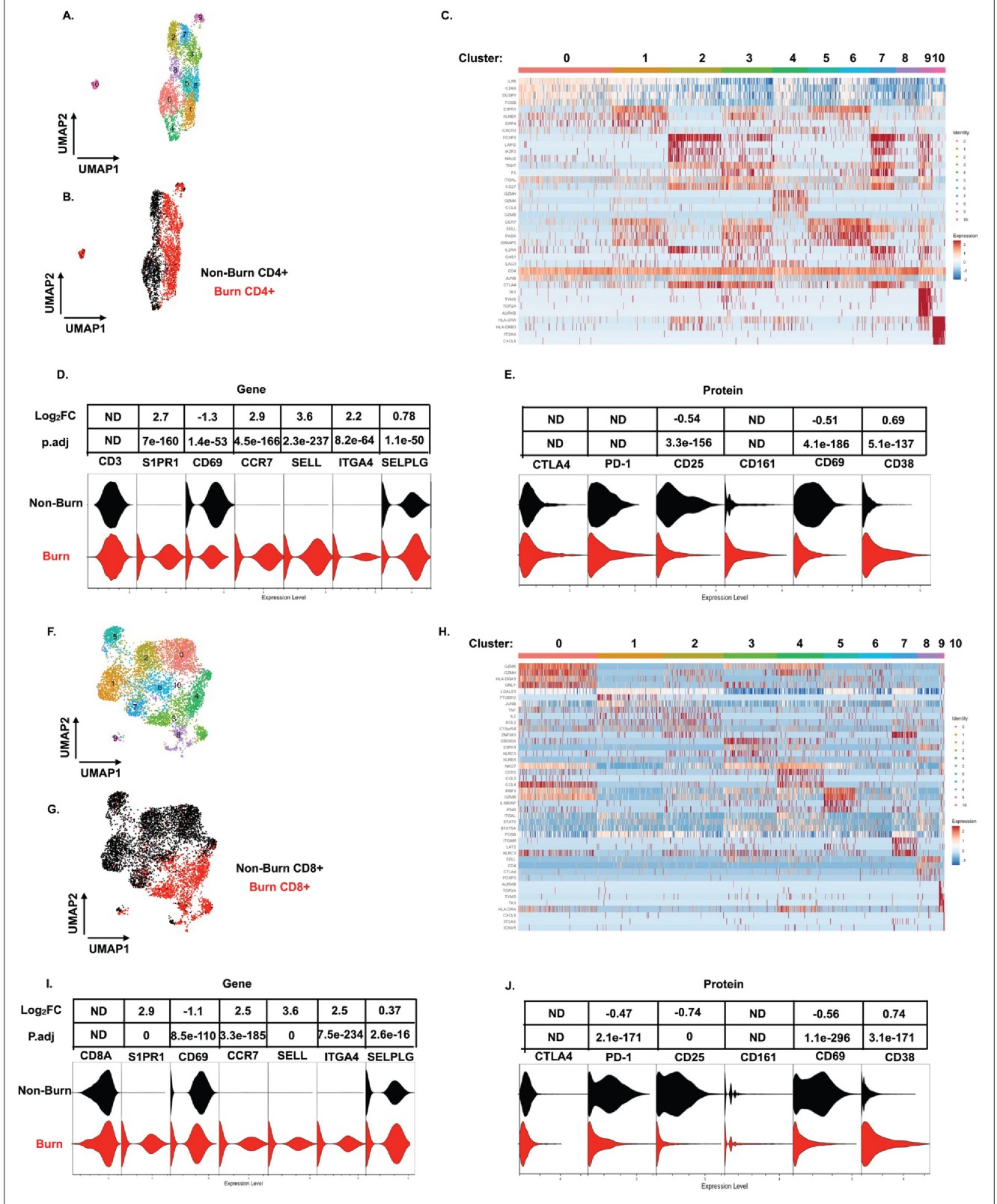

**Figure 5.** scRNA-seq reveals a conventional CD4 + and CD8+ T cell subset that trafficks to burn tissue. Differentially expressed genes define the 10 clusters of CD4 + T cells within burn and non-burn tissue and separate based on tissue origin (**A,B**). Heatmap of the top 8 genes that significantly differentiate CD4 + T cells in burn and non-burn samples (**C**). Violin plot of highly significantly differentially expressed genes relating to tissue residency and homing between burn and non-burn CD4 + T cells (**D**). Violin plot of AbSeq protein expression of receptors between burn and non-burn CD4 +

*Figure 5 continued on next page*

*Figure 5 continued*

T cells (**E**). UMAP analysis define 10 clusters of CD8 + T cells that clustering separately based on tissue origin (**F,G**). Heatmap of the top 8 genes that significantly differentiate CD8 + T cells in burn and non-burn samples (**H**). Violin plot of T cell homing markers and residency markers of CD8 + T cells in burn and non-burn tissue (**I**). Violinplot of AbSeq protein expression of receptors between CD8 + T cells in burn and non-burn tissue (**J**).

The online version of this article includes the following figure supplement(s) for figure 5:

**Figure supplement 1.** Differential expression of cytokines and effector molecules of conventional CD4 + and CD8+ T cells in burn and non-burn tissues.

percentages of IFN-γ+ conventional T cells 7 days post-burn (*Rani et al., 2014*). In total, our observations of conventional T cell phenotypes following burn injury in human skin tissue were similar to those reported in mice, but differences in trafficking of conventional T cells to the burn wound site between mice and humans remains to be determined.

Prior studies of tissue-resident CD4 + T cells in skin of healthy individuals show that they are predominantly CCR7-,( *Boniface et al., 2018*), and more likely to be CD69 + and CCR7- compared with CD4 + T cells in the blood (*Li et al., 2016*). Chronic inflammatory human skin conditions such as psoriasis show skin CD8 + and CD4+CD103+ $T_{RM}$ having no difference to CD103- T cells in IFN-γ production following PMA-ionomycin stimulation (*Kurihara et al., 2019*). We found that in burn tissue, T cells are more likely to be CD69- compared to non-burn tissue, and our single-cell transcriptomics analysis identified a population of CD69CD4 + T cells in burn tissue, which express high levels of CCR7, SELL, S1PR1, receptors not known to be expressed on $T_{RM}$ cells (*Schenkel and Masopust, 2014*; *Kumar et al., 2017*). We hypothesize that following burn injury, there is a migration of peripheral blood CD69CCR7 +CD4+ T cells to the burn site, though these cells are unlikely to contribute to the pro-inflammatory cytokine pool (*Klicznik, 2019*). It is also possible that this lack of CD69 in skin-infiltrating T cells may cause decreased regulation of T cell inflammatory responses due to lower numbers of CD69$^+$ $T_{reg}$ cells (*Gorabi et al., 2020*; *González-Amaro et al., 2013*) Downregulation of CD69 in mice has shown to increase Th17 differentiation and subsequent increase pro-inflammatory cytokines. Consistent with this, we saw a positive association of Th17 differentiation in our KEGG pathway analysis with burn tissue expression of RORC, a transcription factor associated with Th17 cells (*González-Amaro et al., 2013*; *Martín et al., 2010*).

Recent studies have revealed that MAIT cells have the capacity for tissue repair, (*Constantinides et al., 2019*; *Leng et al., 2019*),associated with the expression of IL-17A and related pathways, and prior work in psoriatic skin lesions showed high IL-17A and IL-22 expression though comparison to healthy skin was lacking (*Teunissen et al., 2014*). We did not see significant differences in IL-17A in skin MAIT cells after burn injury, (*Constantinides et al., 2019*), with both non-burn and burn skin samples producing similar IL-17A following PMA-Ionomycin stimulation, at similar levels to that seen by others in psoriatic skin lesions (*Teunissen et al., 2014*). The role of IL-17A+ MAIT cells in tissue repair following burn injury remains unclear. On the other hand, we see a significantly higher percentage of IFN-γ+ and TNF-α+ MAIT cells following PMA-ionomycin treatment in burn tissue compared to non-burn tissue. This data, paired with the large population of MAIT cells expressing cytotoxicity related genes in burn tissue, suggests that MAIT cells may act as a first responder to infection with potential to release cytolytic molecules to respond against infected or damaged cells.

Our study had several limitations. First, due to difficulty with obtaining burn tissue samples, this exploratory study had a low sample size. Thus, we were unable to stratify phenotypes based on sex and age, which are known to impact the frequency and functionality of MAIT cells in particular (*Novak et al., 2014*; *Loh et al., 2020*). Secondly, our use of a targeted approach for scRNA-seq limited the number of genes examined. One cytokine not present in our panel, IL-10, an anti-inflammatory cytokine shown by flow cytometry to have lower expression in burn tissue γδ T cells compared to non-burn tissue, could partly explain the higher pro-inflammatory cytokine production in burn tissue (*Akdis and Blaser, 2001*; *Smith et al., 2018*; *Ng et al., 2013*). Third, our skin samples were collected from several different regions across the body, and skin from different regions may have intrinsic differences in chemokine and cytokine expression (*Béke et al., 2018*). Lastly, the difference in thickness of the dermis layer between burn tissue samples and some non-burn tissue samples could lead to variations in T cell frequency, receptor expression, and cytokine expression, especially in CD69 +CD103+ $T_{RM}$ cells (*Ho and Kupper, 2019*; *Kortekaas Krohn et al., 2022*).

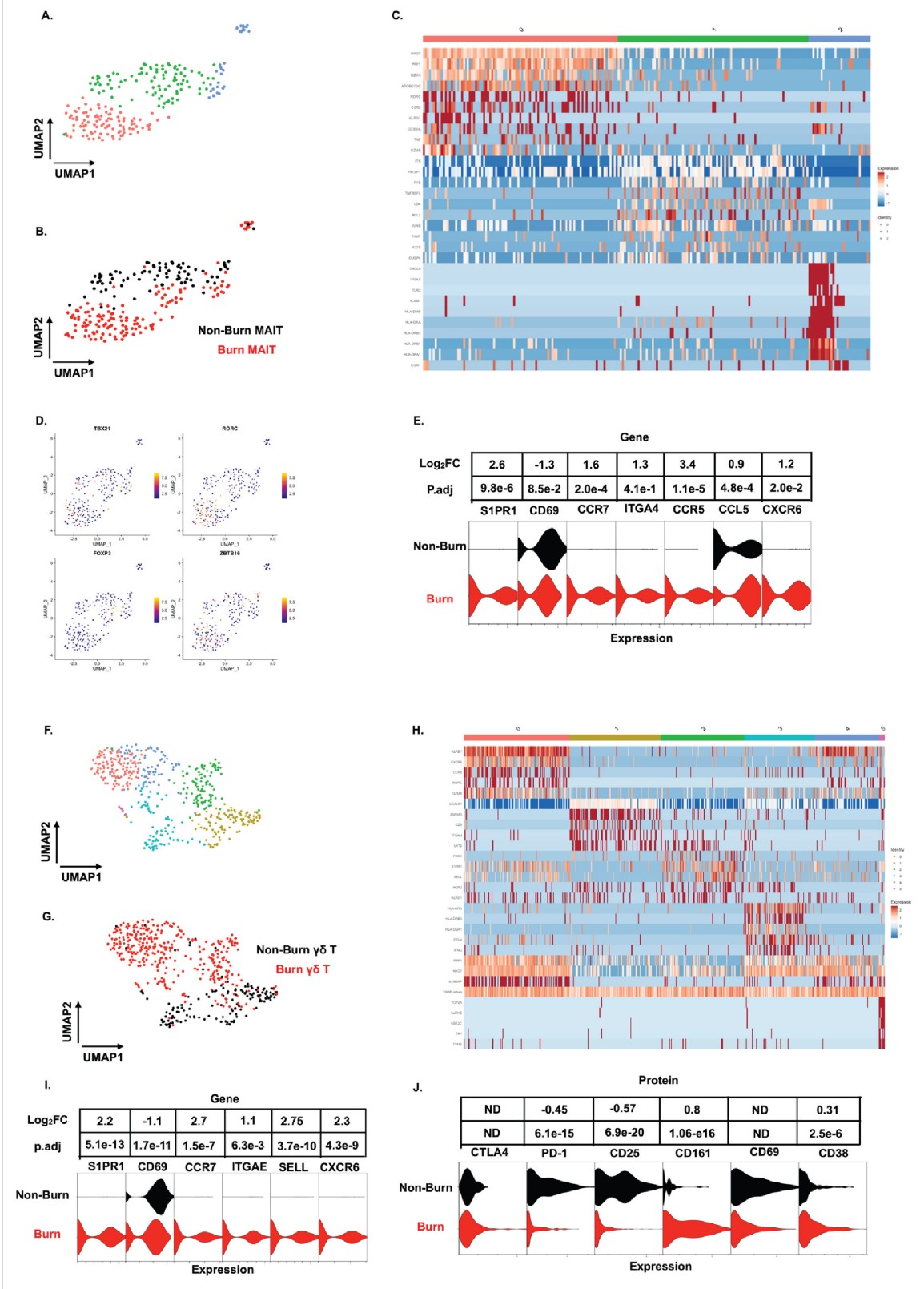

**Figure 6.** Unconventional T cells in burn tissue comprise a highly cytotoxic population not seen in non-burn tissue. UMAP analysis of MAIT cells (defined from AbSeq as Vα7.2+CD161+) within burn and non-burn tissue show three clusters of MAIT cells and their tissue origin (**A,B**). Heatmap of the three clusters of MAIT cells and the top 8 genes that are differentially expressed that define the clusters (**C**). Feature plot showing some commonly expressed MAIT cell transcription factors based on the UMAP in A. (**D**). Violin plot showing significantly differentially expressed T cell homing and residency genes

*Figure 6 continued on next page*

*Figure 6 continued*

between burn and non-burn MAIT cells (**E**). UMAP analysis of γδ T cells (Defined as TRDC +TRGC+ ) reveals 5 separate clusters within burn and non-burn tissue which are distinct based on tissue origin (**F,G**). Heatmap showing the top 5 genes identifying each cluster in F (**H**). Violin plots of significantly differentially expressed T cell homing and residency genes between burn and non-burn γδ T cells (**I**). Violin plots of AbSeq protein expression between burn and non-burn γδ T cells (**J**).

In conclusion, we used flow cytometry, scRNA-seq, and AbSeq methods to show that conventional and unconventional T cell populations in burn skin and non-burn skin are significantly different. T cells in burn skin show higher pro-inflammatory cytokine potential, and the unconventional T cells in burn skin show higher cytotoxic gene expression than non-burn counterparts. Our transcriptomic analysis revealed many differentially expressed genes between burn and non-burn tissue, which could provide information to further investigations on the immune mechanism behind burn injury and wound healing. Overall, the landscape of T cells in burn injury compared to non-burn skin suggests that T cells are shifted towards a pro-inflammatory rather than a homeostatic tissue-resident phenotype. Further investigation of the role that T cells play in tissue repair following burn injury is needed.

## Methods
### Human subjects and enzymatic processing of skin tissue
Samples of discarded non-burn (7 donors) and burn tissue (11 donors) were obtained from through the University of Utah Burn Center, under a protocol (IRB#150686) reviewed by the University of Utah IRB and determined to be non-human subjects research. Discarded tissue was placed into RPMI-1640 media immediately after removal in the surgical suite and brought to laboratory for immediate processing. The skin was washed 1 x with RPMI-1640 and then stamped 16 times with a 3 mm biopsy punch. Four 3 mm pieces were placed in a gentleMACS C tube (Miltenyi Biotech) with 500 µL of RPMI-1640. For a 2 hr digestion, the Tumor Dissociation Kit was used (Miltenyi Biotech) with 100 µL of enzyme M, 25 µL of enzyme R, and 12.5 µL of enzyme A was added to each tube. After digestion 500 µL of cold RPMI-1640 was added and the tubes were placed on a gentleMACS dissociator (Miltenyi Biotech) and set to the h_skin_01 program at room temperature to shear the samples. The tubes were briefly spun down to get any remaining samples and washed with RPMI-1640 twice through a 70 µM strainer. The cells were frozen at –80 °C for no more than 2 months.

### Flow cytometry
For phenotypic analysis of skin mononuclear cells, samples were thawed from –80 °C and treated with human Fc block (BD Biosciences, Cat#564220) for 20 minutes and were stained for surface markers: Zombie UV fixable viability dye (Biolegend, Cat#423107), anti-CD3-BUV395 (BD Biosciences, Cat# 563546), anti-CD8-PE-Cy5.5 (Molecular Probes, Cat#MHCD0818), anti-CD4-BUV496 (BD Biosciences, Cat# 612936), anti-Vα7.2-BV711 (Biolegend, Cat# 351731), anti-LAG3-BV785 (Biolegend, Cat# 369321), anti-CD69-BUV563 (BD Biosciences, Cat#748764), anti-TCRγδ-BV480 (BD Biosciences, Cat# 566076), anti-TCR Vα24-Jα18-APC-Fire 750 (Biolegend, Cat# 342927), anti-CTLA-4-PE-Cy5 (BD Biosciences, Cat#555854), anti-CD103-PE-Cy7 (Biolegend, Cat#350211), anti-CD25-BV650 (Biolegend, Cat# 02037), anti-PD-1-BV605 (Biolegend, Cat# 329923), anti-CD161-BV605 (Biolegend, Cat#339915), anti-TIM3-BV421 (Biolegend, Cat# 345007), anti-CD38-APC-Fire-810 (Biolegend, Cat# 303549), and anti-human MR1 5-OP-RU Tetramer (NIH Tetramer Core Facility). Skin mononuclear cells were stimulated with PMA-Ionomycin and extracellular transport blocked by brefeldin A for 3hr were stained for surface markers: anti-human MR1-5-OP-RU Tetramer (NIH Tetramer Core Facility), anti-CD3-BUV395 (BD Biosciences, Cat# 563546), anti-CD8-PE-Cy5.5 (Molecular Probes, Cat#MHCD0818), anti-CD4-BUV496 (BD Biosciences, Cat# 612936), anti-Vα7.2-BV711 (Biolegend, Cat# 351731), anti-TCRγδ-BV480 (BD Biosciences, Cat# 566076), anti-TCR Vα24-Jα18-APC-Fire 750 (Biolegend, Cat# 342927) and intracellularly stained for: anti-IL-17A-BV650 (BD Biosciences, Cat# 563746), anti-IL-10-PerCp-Cy5.5 (Biolegend, Cat# 501417), anti-IL-6-FITC (Biolegend, Cat# 501103), anti-IFN-γ-Alexa Fluor 700 (Biolegend, Cat#506515), anti-TNFα-ef450 (Invitrogen, Cat#48-7349-42). Sample data were acquired using a 5-laser Cytek Aurora flow cytometer (Cytek) and analyzed using FlowJo software v10 (Tree Star, Inc Ashland, OR).

## Targeted T cell single-cell mRNA sequencing and analysis

Three samples each of discarded non-burn and burn tissue were processed as previously stated. The skin mononuclear cells were treated with human Fc block (BD Biosciences, Cat#564220) for 20 minutes and were stained with eFlour 780 fixable viability dye (Invitrogen, Cat# 65-2860-40) and anti-CD3-Alexa Fluor 700 (Biolegend, Cat#300423) and CD3 +T cells were sorted on a 4-laser BD FACSAria 3. Cells were incubated with BD AbSeq Ab-oligo V$\alpha$7.2 and CD161 to identify MAIT cells within the CD3 +T cell populations. Single cells were isolated using Single Cell Capture system (BD Biosciences) and analyzed for live cells and amount of total single cells captured with the BD Rhapsody Single-cell Analysis System. Sequencing libraries of mRNA and cDNA of AbSeq were doing simultaneously using BD Rhapsody targeted mRNA and AbSeq amplification kits and protocol. The final libraries of the AbSeq and mRNA were analyzed using Agilent 2200 TapeStation. The AbSeq-oligos, Sample Multiplex Tags, and T cell mRNA targeted libraries were pooled together before sequencing on a NovaSeq6000 instrument (Illumina). For sequencing, 12,000 reads/cell were dedicated toward the targeted mRNA library, 10,000 reads/AbSeq-oligo, and 200 reads/Sample Tag. The FASTQ files were uploaded to Seven Bridges Genomics, and a workflow designed by BD Biosciences was used to analyze the data to demultiplex, identify cells based on AbSeq-oligo, and analyze the single-cell mRNA data. For specific instructions, refer to BD Single Cell Genomics Bioinformatics Handbook, Doc ID: 54169, Rev. 6.0 (*Mair et al., 2020*). Analysis of the final count matrix was done with Seurat v4.0.3 in R v4.1.2. Using the Seurat object, gene ontology (database: GO_Molecular_Function_2021) and KEGG (database: KEGG_2021_Human) were performed through enrichR v3.1 package in R v4.1.2.

## Statistical analysis

For comparisons of non-burn tissue with burn tissue flow cytometric data, a Mann-Whitney *U* test was used. GraphPad Prism 8.3.0 software was used for statistical analysis of the flow cytometry data and $p < 0.05$ was considered statistically significant. For comparisons of scRNA-seq data between burn and non-burn, Seurat function FindMarkers with test MAST was used with min.pct=0.10 (*Finak et al., 2015*). Volcano plots showing differentially expressed genes between burn and non-burn T cells were generated using EnhancedVolcano v1.12.0 package in R v4.1.2.

## Datasets Generated

scRNA-seq of sorted human CD3+ T cells from discarded burn and non-burn skin: Daniel Labuz and Daniel Leung, 2021, https://www.ncbi.nlm.nih.gov/sra/SRR23092707, SRA, SRR23092707.

scRNA-seq of sorted human CD3+ T cells from discarded burn and non-burn skin: Daniel Labuz and Daniel Leung, 2021, https://www.ncbi.nlm.nih.gov/sra/SRR23092708, SRA, SRR23092708.

## Acknowledgements

This research was supported by the National Institutes of Health (R01AI130378 to DTL and TL1TR002540 to DL). We would like to thank all the study subjects who participated in the study. We would like to thank all the nurses and caregivers at the University of Utah burn center. We would also like to thank the staff of the University of Utah Flow Cytometry Core.

## Additional information

### Funding

| Funder | Grant reference number | Author |
|---|---|---|
| National Institutes of Health | R01AI130378 | Daniel T Leung |
| National Institutes of Health | TL1TR002540 | Daniel R Labuz |

The funders had no role in study design, data collection and interpretation, or the decision to submit the work for publication.

## Author contributions
Daniel R Labuz, Conceptualization, Data curation, Software, Formal analysis, Validation, Investigation, Visualization, Methodology, Writing - original draft, Project administration, Writing - review and editing; Giavonni Lewis, Conceptualization, Resources, Data curation, Supervision, Validation, Project administration, Writing - review and editing; Irma D Fleming, Callie M Thompson, Resources; Yan Zhai, Resources, Data curation; Matthew A Firpo, Conceptualization, Resources, Validation, Writing - review and editing; Daniel T Leung, Conceptualization, Supervision, Funding acquisition, Methodology, Writing - original draft, Project administration, Writing - review and editing

## Author ORCIDs
Daniel R Labuz ⓘ http://orcid.org/0000-0003-3769-3544
Daniel T Leung ⓘ http://orcid.org/0000-0001-8401-0801

## Ethics
Human subjects: The study protocol (IRB#150686) was reviewed by the University of Utah IRB and determined to be non-human subjects research.

## Decision letter and Author response
Decision letter https://doi.org/10.7554/eLife.82626.sa1
Author response https://doi.org/10.7554/eLife.82626.sa2

## Additional files

### Supplementary files
• MDAR checklist

### Data availability
Files containing the analysis code is available at https://github.com/LeungLab/MAIT-Burn-Injury (copy archived at swh:1:rev:55d8c79ead016cff2a8fef97e94c381ae82f5f34 ). The scRNA sequencing files have been uploaded to NCBI BioProject PRJNA924066 (NCBI SRA. ID: SRR23092708 and SRR23092707).

The following datasets were generated:

| Author(s) | Year | Dataset title | Dataset URL | Database and Identifier |
|---|---|---|---|---|
| Labuz D, Leung D | 2023 | scRNA-seq of sorted human CD3+ T cells from discarded burn and non-burn skin | https://www.ncbi.nlm.nih.gov/sra/SRR23092707 | NCBI SRA, SRR23092707 |
| Labuz D, Leung D | 2023 | scRNA-seq of sorted human CD3+ T cells from discarded burn and non-burn skin | https://www.ncbi.nlm.nih.gov/sra/SRR23092708 | NCBI SRA, SRR23092708 |

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
