## [Editor Report]

The work in analyzing the T cell repertoire by multi-omics analysis is very valuable for the field of wound biology and provides convincing data with regard to both conventional and conventional T cells and their putative contributions. This moves the field beyond examining classical mediators of wound healing such as macrophages and neutrophils. We look forward to seeing this important work in *eLife*.

---

## [Decision Letter]

**Decision letter after peer review:**

Thank you for submitting your article "Targeted multi-omic analysis of human skin tissue identifies alterations of conventional and unconventional T cells associated with burn injury" for consideration by *eLife*. Your article has been reviewed by 2 peer reviewers, and the evaluation has been overseen by a Reviewing Editor and Mone Zaidi as the Senior Editor. The following individual involved in the review of your submission has agreed to reveal their identity: Lucas F. de Andrade (Reviewer #1).

Essential revisions:

1) Can the authors give a more quantitative assessment of cytokine levels from the scRNA-seq datasets?

2) Please give both percentage and number of cells/mm3 for CD3+ and the subpopulations of T cells.

3) Can GSEA analyses be done to explain what pathways are actually enriched?

*Reviewer #1 (Recommendations for the authors):*

The authors should show the absolute numbers of each one of the T cell populations. That is done for total T cells, but I am asking about each one of them. The importance of that analysis is that when considering % a population decreases and another augments. Therefore it is hard to determine if there are more CD4 T cells or if their % just looks higher because there are fewer CD8 T cells, for example.

The authors could attempt to perform a gene set enrichment analysis with the sc RNA seq data to perhaps gain more information about additional functions of the T cell populations.

CD69 expression is highly different when comparing T cells from burned to non-burned skins. While we use it as a tissue residency marker, it also plays a function in lymphocytes. For example, CD69 appears to activate NK cells (PMID: 10447727). The authors could comment if the lack of CD69 alters the functions of skin-infiltrating T cells.

*Reviewer #2 (Recommendations for the authors):*

The work is very interesting with the caveats below.

– Figure 1. The authors calculate both percentage and number of cells/mm3 for CD3+, while for the subpopulations only the percentage is presented. Because percentages can be misleading, authors should also present the numbers/mm3 for the other populations (even if only as Supplementary Material).

– The authors should include an example of the complete sequence of the gating strategy used in their flow cytometry (e.g. from FSCxSSC to the specific markers). This should be included in the Supplementary material.

– In the first part of the manuscript, there is an emphasis on the pro-inflammatory state of T cells in the burned skin tissue samples, which show enhanced production of cytokines like TNFa and IFNγ. This is not investigated or discussed, however, when the results from scRNA-seq are presented. What happens to the expression levels of these and other cytokines (e.g. chemokines)? While the heatmap shows cluster markers, it would be good to have the cytokine expression levels shown as well and whether these are down- or upregulated after burn injury. This could be done by creating volcano plots where each T cell subset is compared (normal vs burn tissue samples), with specific cytokines indicated as well as the threshold of significance (p-value). Authors should take advantage of their rich dataset to present information about any relevant cytokines there (including chemokines) and not be limited to the ones analyzed by flow cytometry.

– Although this is a descriptive work, the authors could explore more the meaning of their scRNA-seq data. A standard way to get an overall idea of which cellular processes and pathways are being stimulated or inhibited in large datasets like RNA-seq is the use of tools for pathway enrichment analysis. Therefore, authors should use their lists of genes affected by burn injury to perform a broader and unbiased analysis of how the function of these T cells is changing and how this may be related to processes like wound healing.

---

## [Author Response]

Essential revisions:1) Can the authors give a more quantitative assessment of cytokine levels from the scRNA-seq datasets?

We have added two supplemental figures showing the expression of cytokines by scRNA-seq in CD3+, CD8^+^ and CD4^+^ T cells. These are Figure 4-Supplement Figure 1 and Figure 5-Supplement Figure 1

Lines 212-214:

“Lastly, we see that expression of effector molecules IFNΓ, TNF, NAMPT, GZMB, GZMH, and PRF1 is lower in burn compared to non-burn CD3+ T cells (Figure 4-Supplement Figure 1A-C).”

Lines 233-236:

“A closer look at effector molecules shows that burn CD4^+^ T cells have lower expression of *NAMPT*, *GZMA*, *PRF1*, *TNF*, and *CCL5* compared to non-burn CD4^+^ T cells (Figure 5-Supplement Figure 1A-B).”

Lines 241-243:

“We also saw that expression of *NAMPT, TNF, IFNΓ, CCL4, and IL32* were significantly lower in burn tissue CD8^+^ T cells compared to non-burn CD8^+^ T cells (Figure 5-Supplement Figure 1C-D)

2) Please give both percentage and number of cells/mm3 for CD3+ and the subpopulations of T cells.

We have added a supplemental figure (Figure 1-Supplement Figure 2) that includes cell per mm^3^ for CD4 and CD8 T cell populations.

Lines 125-127:

“We did not see any significant differences in absolute numbers of CD4^+^ or CD8^+^ T cells between burn and non-burn skin. (Figure 1-Supplement Figure 2B-C).”

3) Can GSEA analyses be done to explain what pathways are actually enriched?

We have added a supplemental figure that look at gene ontology (GO) and KEGG analyses for CD3 T cell scRNA-seq data, Figure 4-Supplement Figure 1.

Lines 214-221:

“Gene ontology (GO) analysis shows that T cell activation and positive regulation of cytokine production are associated with burn T cells while apoptotic processes are negatively associated with burn T cells. (Figure 4-Supplement Figure 1D). Lastly, KEGG analysis of CD3+ T cells shows Th17, Th1, and Th2 cell differentiation are highly associated with burn T cells while cytokine-cytokine receptor interactions and T cell receptor signaling pathways are downregulated in burn T cells (Figure 4-Supplement Figure 1E).”

We chose not to use gene set enrichment analysis (GSEA) due to the experiment being a targeted scRNA-seq approach. In total, there were 256 genes following quality control that were captured in our experiment. Compared with GSEA analysis, GO and KEGG pathway analyses were more informative and identified enriched pathways that were more consistent with what is known in the literature.

Reviewer #1 (Recommendations for the authors):The authors should show the absolute numbers of each one of the T cell populations. That is done for total T cells, but I am asking about each one of them. The importance of that analysis is that when considering % a population decreases and another augments. Therefore it is hard to determine if there are more CD4 T cells or if their % just looks higher because there are fewer CD8 T cells, for example.

As addressed above under for Essential revisions 2:

We have added a supplemental figure (Figure 1-Supplement Figure 2) that includes cell per mm^3^ for CD4 and CD8 T cell populations.

Lines 125-127:

“We did not see any significant differences in absolute numbers of CD4^+^ or CD8^+^ T cells between burn and non-burn skin. (Figure 1-Supplement Figure 2B-C).”

The authors could attempt to perform a gene set enrichment analysis with the sc RNA seq data to perhaps gain more information about additional functions of the T cell populations.

As addressed above under for Essential revisions 3:

We have added a supplemental figure that look at gene ontology (GO) and KEGG analyses for CD3 T cell scRNA-seq data, Figure 4-Supplement Figure 1.

Lines 214-221:

“Gene ontology (GO) analysis shows that T cell activation and positive regulation of cytokine production are associated with burn T cells while apoptotic processes are negatively associated with burn T cells. (Figure 4-Supplement Figure 1D). Lastly, KEGG analysis of CD3+ T cells shows Th17, Th1, and Th2 cell differentiation are highly associated with burn T cells while cytokine-cytokine receptor interactions and T cell receptor signaling pathways are downregulated in burn T cells (Figure 4-Supplement Figure 1E).”

We chose not to use gene set enrichment analysis (GSEA) due to the experiment being a targeted scRNA-seq approach. In total, there were 256 genes following quality control that were captured in our experiment. Compared with GSEA analysis, GO and KEGG pathway analyses were more informative and identified enriched pathways that were more consistent with what is known in the literature.

CD69 expression is highly different when comparing T cells from burned to non-burned skins. While we use it as a tissue residency marker, it also plays a function in lymphocytes. For example, CD69 appears to activate NK cells (PMID: 10447727). The authors could comment if the lack of CD69 alters the functions of skin-infiltrating T cells.

We added a section in discussion talking about relevancy of CD69.

Lines 337-343:

“It is also possible that this lack of CD69 in skin-infiltrating T cells may cause decreased regulation of T cell inflammatory responses due to fewer CD69+ Treg cells.^49,50^ Downregulation of CD69 in mice has shown to increase Th17 differentiation and subsequent increase pro-inflammatory cytokines. Consistent with this, we saw a positive association of Th17 differentiation in our KEGG pathway analysis with burn tissue expression of RORC, a transcription factor associated with Th17 cells.^50,51^”

Reviewer #2 (Recommendations for the authors):The work is very interesting with the caveats below.– Figure 1. The authors calculate both percentage and number of cells/mm3 for CD3+, while for the subpopulations only the percentage is presented. Because percentages can be misleading, authors should also present the numbers/mm3 for the other populations (even if only as Supplementary Material).

As addressed above for reviewer 1:

We have added a supplemental figure (Figure 1-Supplement Figure 2) that includes cell per mm^3^ for CD4 and CD8 T cell populations.

Lines 125-127:

“We did not see any significant difference in amount of CD4^+^ or CD8^+^ T cells between burn and non-burn skin. (Figure 1-Supplement Figure 2B-C).”

– The authors should include an example of the complete sequence of the gating strategy used in their flow cytometry (e.g. from FSCxSSC to the specific markers). This should be included in the Supplementary material.

We have added the gating strategy for all the T cells from FSCxSSC in Figure 1-Supplement Figure 2.

– In the first part of the manuscript, there is an emphasis on the pro-inflammatory state of T cells in the burned skin tissue samples, which show enhanced production of cytokines like TNFa and IFNγ. This is not investigated or discussed, however, when the results from scRNA-seq are presented. What happens to the expression levels of these and other cytokines (e.g. chemokines)? While the heatmap shows cluster markers, it would be good to have the cytokine expression levels shown as well and whether these are down- or upregulated after burn injury. This could be done by creating volcano plots where each T cell subset is compared (normal vs burn tissue samples), with specific cytokines indicated as well as the threshold of significance (p-value). Authors should take advantage of their rich dataset to present information about any relevant cytokines there (including chemokines) and not be limited to the ones analyzed by flow cytometry.

As addressed above for Essential Revisions 1:

We have added two supplemental figures showing the expression of cytokines by scRNA-seq in CD3+, CD8^+^ and CD4^+^ T cells. These are Figure 4-Supplement Figure 1 and Figure 5-Supplement Figure 1.

Lines 212-214:

“Lastly, we see that expression of effector molecules IFNΓ, TNF, NAMPT, GZMB, GZMH, and PRF1 is lower in burn compared to non-burn CD3+ T cells (Figure 4-Supplement Figure 1A-C).”

Lines 233-236:

“A closer look at effector molecules shows that burn CD4^+^ T cells have lower expression of *NAMPT*, *GZMA*, *PRF1*, *TNF*, and *CCL5* compared to non-burn CD4^+^ T cells (Figure 5-Supplement Figure 1A-B).”

Lines 241-243:

“We also saw that expression of *NAMPT, TNF, IFNΓ, CCL4, and IL32* were significantly lower in burn tissue CD8^+^ T cells compared to non-burn CD8^+^ T cells (Figure 5-Supplement Figure 1C-D)

– Although this is a descriptive work, the authors could explore more the meaning of their scRNA-seq data. A standard way to get an overall idea of which cellular processes and pathways are being stimulated or inhibited in large datasets like RNA-seq is the use of tools for pathway enrichment analysis. Therefore, authors should use their lists of genes affected by burn injury to perform a broader and unbiased analysis of how the function of these T cells is changing and how this may be related to processes like wound healing.

As addressed above under for Essential revisions 3:

We have added a supplemental figure that look at gene ontology (GO) and KEGG analyses for CD3 T cell scRNA-seq data, Figure 4-Supplement Figure 1.

Lines 214-221:

“Gene ontology (GO) analysis shows that T cell activation and positive regulation of cytokine production are associated with burn T cells while apoptotic processes are negatively associated with burn T cells. (Figure 4-Supplement Figure 1D). Lastly, KEGG analysis of CD3+ T cells shows Th17, Th1, and Th2 cell differentiation are highly associated with burn T cells while cytokine-cytokine receptor interactions and T cell receptor signaling pathways are downregulated in burn T cells (Figure 4-Supplement Figure 1E).”

We chose not to use gene set enrichment analysis (GSEA) due to the experiment being a targeted scRNA-seq approach. In total, there were 256 genes following quality control that were captured in our experiment. Compared with GSEA analysis, GO and KEGG pathway analyses were more informative and identified enriched pathways that were more consistent with what is known in the literature.